# Upregulation of Thioredoxin Reductase 1 Expression by Flavan-3-Ols Protects Human Kidney Proximal Tubular Cells from Hypoxia-Induced Cell Death

**DOI:** 10.3390/antiox11071399

**Published:** 2022-07-19

**Authors:** Jixiao Zhu, Manqin Fu, Jian Gao, Guoyu Dai, Qiunong Guan, Caigan Du

**Affiliations:** 1Department of Urologic Sciences, University of British Columbia, Vancouver, BC V5Z 1M9, Canada; 20081024@jxutcm.edu.cn (J.Z.); funanqin@gdaas.cn (M.F.); jgao@prostatecentre.com (J.G.); gdai@prostatecentre.com (G.D.); qiunong.guan@ubc.ca (Q.G.); 2Research Center for Traditional Chinese Medicine Resources and Ethnic Minority Medicine, Jiangxi University of Traditional Chinese Medicine, Nanchang 330004, China; 3Sericultural & Agri-Food Research Institute, Guangdong Academy of Agricultural Sciences/Key Laboratory of Functional Foods, Ministry of Agriculture/Guangdong Key Laboratory of Agricultural Products Processing, Guangzhou 510610, China

**Keywords:** hypoxia, oxidative stress, flavan-3-ols, renal tubular epithelial cells, TXNRD1

## Abstract

Renal hypoxia and its associated oxidative stress is a common pathway for the development of kidney diseases, and using dietary antioxidants such as flavan-3-ols to prevent kidney failure has received much attention. This study investigates the molecular mechanism by which flavan-3-ols prevent hypoxia-induced cell death in renal tubular epithelial cells. Human kidney proximal tubular cells (HKC-8) were exposed to hypoxia (1% O_2_) in the presence of flavan-3-ols (catechin, epicatechin, procyanidin B1, and procyanidin B2). Cell death was examined using flow cytometric analysis. Gene expression was determined using a PCR array and Western blotting, and its network and functions were investigated using STRING databases. Here, we show that the cytoprotective activity of catechin was the highest among these flavan-3-ols against hypoxia-induced cell death in cultured HKC-8 cells. Exposure of HKC-8 cells to hypoxia induced oxidative stress leading to up-regulation of *DUOX2*, *NOX4*, *CYBB* and *PTGS2* and down-regulation of *TXNRD1* and *HSP90AA1*. Treatment with catechin or other flavan-3-ols prevented the down-regulation of *TXNRD1* expression in hypoxic HKC-8 cells. Overexpression of TXNRD1 prevented hypoxia-induced cell death, and inactivation of TXNRD1 with TRi-1, a specific TXNRD1 inhibitor, reduced the catechin cytoprotection against hypoxia-induced HKC-8 cell death. In conclusion, flavan-3-ols prevent hypoxia-induced cell death in human proximal tubular epithelial cells, which might be mediated by their maintenance of TXNRD1 expression, suggesting that enhancing TXNRD1 expression or activity may become a novel therapeutic strategy to prevent hypoxia-induced kidney damage.

## 1. Introduction

Hypoxia is a state of inadequate oxygen (O_2_) during which normal cellular homeostasis in the tissue cannot be maintained. In the kidney, there are many mechanisms by which O_2_ diffusion and supply to the tubular and interstitial cells are impaired such as glomerular injury, loss of peritubular capillaries, hemodynamic maladjustment, repeated acute kidney injury, and systemic anemia [1,2]. The chronic hypoxia within the kidney aggravates oxidative stress and induces a sustained inflammation including infiltrating leukocytes [2,3], which consequently leads to the development of chronic tubulointerstitial injury or chronic kidney disease (CKD), and as a key factor the chronic hypoxia, drives the CKD progression from early stages to end-stage renal failure [1,2,4]. Thus, therapeutic targeting of the chronic hypoxia is a promising strategy against various forms of kidney disease.

Flavan-3-ols (flavanols) are a group of natural polyphenolic compounds containing a chemical structure of 2-phenyl-3,4-dihydro-2H-chromen-3-ol, and they are widely found in various food products such as fruits and vegetables [5,6]. These chemicals range from the single-molecular (monomer) catechin and epicatechin to oligomers such as oligomeric proanthocyanidins [7,8]. There has been an increasing interest in the potential of flavan-3-ols for their anti-inflammatory and antioxidant activities including modulation of cell redox status and disruption of cellular signaling pathways such as NF-κB [7]. In *Vitis vinifera* (grape), the major flavan-3-ols or catechins are catechin, epicatechin and procyanidin dimers B1 (PCB1) and B2 (PCB2) [9]. Numerous experimental studies have demonstrated the protective effect of grape extract or catechins against kidney injury in animal models of kidney ischemia-reperfusion and CKD [10,11,12,13]. Further, a clinical trial has reported that supplementation with grape seed extract (2 g/day) has improved some kidney function parameters such as glomerular filtration rate and proteinuria in CKD patients (stage 2–4) [14]. The objective of this study was to investigate the molecular actions of the common catechins or flavan-3-ols of grape in the hypoxia-mediated cell death of cultured human renal tubular epithelial cells.

## 2. Materials and Methods

### 2.1. Cell Cultures and Reagents

Human proximal tubular epithelial cell line HKC-8 was kindly provided by Dr. Daniel L. Sparks (Ottawa, ON, Canada) with permission from Dr. Lorraine Racusen [15] and were grown in complete K1^+/+^ medium [16]. The HKC-8 cells were converted to a TXNRD1-overexpressing cell type (HKC-8^TXNRD1^) by stable expression of pHEX6300 vector containing human TXNRD1 cDNA, a gift from Dr. Nicola Burgess-Brown (University of Oxford, UK) (Addgene plasmid #38863), and HKC-8 cells with stable expression of the empty pHEX6300 vector were used as a control cell type (HKC-8^Control^). Both types of cells were grown and maintained in K1^+/+^ medium [Dulbecco’s modified Eagle’s medium (DMEM):Hams F12] (50:50), supplemented with 5% bovine calf serum, hormone mixture (5 µg/mL insulin, 1.25 ng/mL prostaglandin E1, 34 pg/mL triiodothyronine, 5 µg/mL transferrin, 1.73 ng/mL sodium selenite, and 18 ng/mL hydrocortisone) and 25 ng/mL epidermal growth factor] in the presence of the selective antibiotic zeocin (up to 100 μg/mL) [16]. The overexpression of TXNRD1 protein in HKC-8^TXNRD1^ cells compared with that in HKC-8^Control^ cells was confirmed using Western blot analysis (Appendix A).

Pure (+)-catechin and (−)-epicatechin (both ≥98% of purity) were purchased from Sigma-Aldrich Canada (Oakville, ON, Canada), and PCB1 and PCB2 (both ≥98% of purity) from Cayman Chemical (Ann Arbor, MI, USA, and TRi-1 from MedChemExpress (MCE) (distributor: Cedarlane Labs, Burlington, ON, Canada). The chemical structure of the catechin, epicatechin, PCB1 and PCB2 is illustrated in Appendix A, and their 1,1-diphenyl-2-picrylhydrazyl (DPPH) radical scavenging activities in Appendix A.

### 2.2. Induction of Hypoxia In Vitro

Hypoxia in cultured HKC-8, HKC-8^TXNRD1^ and HKC-8^Control^ cells in complete K1^+/+^ medium was induced by incubation in a humidified hypoxic chamber (Coy Laboratory Products, Inc., Grass Lake, MI, USA) that was provided with an atmosphere containing 1% O_2_, 5% CO_2_ and 94% N_2_ at 37 °C for 72 h. The normoxia control cell cultures were grown a humidified CO_2_ incubator (5% CO_2_ and 95% air (approximately 20% O_2_)) at 37 °C.

### 2.3. Flow Cytometric Determination of Cell Death, Reactive Oxygen Species, and Mitochondrial Membrane Potential

Cell death was quantitatively determined using fluorescence-activated cell sorter (FACS) analysis with Annexin-V conjugated with phycoerythrin (Annexin-V-PE) and 7-amino-actinomycin D (7-AAD) double staining. 7-AAD is a cell membrane-impermeable dye that specifically stains nucleic acids inside the cell. A positive 7-AAD staining indicates a cell with compromised plasma membranes as seen in necrosis and late apoptosis. Annexin V specifically binds to phosphatylserine that is exclusively found on the cell surface when a cell is undergoing apoptosis. Therefore, the positive staining with Annexin-V-PE represents early and late stages of apoptosis. Briefly, cells were incubated with Annexin-V-PE and 7-AAD for 15 min in the dark. The proportions (%) of total cell death that included early apoptosis (Annexin-V positive, left lower quadrant—Q3), late apoptosis (both Annexin-V-PE and 7-AAD positive, left upper quadrant—Q2) and necrosis (7-AAD positive, upper right quadrant—Q1) were determined compared with unstained controls by a flow cytometry and quantified using FlowJo software (Tree Star Inc., Ashland, OR, USA).

The cellular levels of reactive oxygen species (ROS) were measured using FACS analysis with 2′-7′-Dichlorodihydrofluorescein diacetate (DCF-DA) (Sigma-Aldrich Canada, Oakville, ON, Canada) as described previously [17]. DCF-DA is a cell permeable fluorogenic dye that can be oxidized by hydroxyl, peroxyle and other ROS within the cell into 2′,7′-dichlorodidrofluorescein (green DCF), and the green fluorescence intensity of the resultant DCF is an indicator of intracellular ROS. In brief, HKC-8 cells (after hypoxia exposure or in normoxia) were incubated with 0.125 µM DCF-DA for 30 min in the dark, and the mean fluorescence intensity (MFI) of DCF was measured by flow cytometry and quantified using FlowJo Software (Tree Star Inc., Ashland, OR, USA).

Mitochrondrial dysfunction is commonly determined by the change of mitochondrial membrane potential (MMP) and can be detected by FACS with JC-1 iodide dye (Sigma-Aldrich Canada, Oakville, ON, Canada) [18]. JC-1 is a cationic carbocyanine dye that can diffuse into and accumulate in mitochrondria. At a low level, it exists as a monomer (green) and at a high level it forms red aggregates. Therefore, the accumulation or aggregation of JC-1 dye can be used as a marker of MMP. In brief, HKC-8 cells (after hypoxia exposure or in normoxia) were incubated with 0.2 µM JC-1 iodide for 30 min in the dark, and the red fluorescence (FL-2) of JC-1 aggregation in hypoxia-treated cells was determined compared with that of cells in normoxia by flow cytometry and quantified using FlowJo Software (Tree Star Inc., Ashland, OR, USA).

### 2.4. PCR Array Analysis of Oxidative Stress-Related Gene Expression

The expression of oxidative stress-associated 84 genes was quantitatively examined using PCR Arrays kits following the manufacturer’s instructions (Cat. No. 330231 PAHS 065ZA, SABiosciences—QIAGEN Inc., Valencia, CA, USA) (for detail of the gene list examined, please see Appendix A). Each group (hypoxic HKC-8 versus normoxic HKC-8; catechin-treated hypoxic HKC-8 versus untreated hypoxic HKC-8) consisted of four different samples that were collected at separate times for determination of the gene expression profiles using PCR arrays. The total RNA from each sample was directly extracted and purified from the monolayer after 72 h of hypoxia or in normoxia using the RNeasy Microarray Tissue Mini kit (QIAGEN, Valencia, CA, USA), and was converted to cDNA using RT^2^ First Strand Kit (QIAGEN, Valencia, CA, USA). The expression of selected genes was amplified by real-time PCR using RT^2^ Profile PCR arrays (QIAGEN, Valencia, CA, USA). The data were analyzed using Web-based PCR Array Data Analysis Software (accessed on 14 October 2018; www.SABiosciences.com/pcrarraydataanalysis.php).

### 2.5. Analysis of Protein-Protein Interaction and Functional Association Networks

The interactions and functions of affected oxidative stress-related genes were analyzed using STRING databases [19], and the data were generated following the instructions on the website (accessed on 28 March 2022; https://string-db.org/).

### 2.6. Western Blot Analysis

The protein levels of TXNRD1 or hypoxia-inducible factor 1α (HIF-1α) and housekeeping Glyceraldehyde 3-phosphate dehydrogenase (GAPDH) were determined using Western blot analysis. In brief, whole cellular protein was extracted from cultured cells by a brief sonication in RIPA buffer (50 mM Tris-HCl, pH 8.0, 150 mM NaCl, 1% Nonidet P-40, 0.5% sodium deoxycholate, and 0.1% SDS) containing protease inhibitor cocktail (Roche, Mannheim, Germany), and fractioned by 10% SDS-polyacrylamide gel electrophoresis (SDS-PAGE) for detection of TXNRD1 or 7% SDS-PAGE for HIF-1α. After being transferred to nitrocellulose membranes (Bio-Rad Lab, Hercules, CA, USA), the TXNRD1 proteins (55 kDa) on the blot were identified using rabbit monoclonal anti-TXNRD1 antibody (Ab 124954, Abcam, Toronto, ON, Canada) or HIF-1α (~120 kDa) using mouse monoclonal anti-HIF-1α antibody (sc-53546, Santa Cruz Biotech, Dallas, TX, USA) along with horseradish peroxidase (HRP)-conjugated secondary antibodies. The blots were reprobed using housekeeping GAPDH (36 kDa) with anti-GAPDH antibody (Epitope Biotech Inc., Vancouver, BC, Canada) to confirm equal protein loading in each sample. The expression levels of the TXNRD1 or HIF-1α proteins were measured using a densitometry, and were calculated as the ratio of the TXNRD1 or HIF-1α protein to the GAPDH on the same blots.

### 2.7. Statistical Analysis

Data were collected from individual experiments and were presented as mean ± standard derivation (SD) of each group. The statistical analysis of differences between groups was performed by *t*-test (two-tailed distribution) or analysis of variance (ANOVA) as appropriate with Prism^®^ software (GraphPad Software, Inc., La Jolla, CA, USA). A *p* value of ≤0.05 was considered statistically significant.

## 3. Results

### 3.1. Reduction of Cell Death by Grape-Derived Flavan-3-Ols in Hypoxic HKC-8 Cells

The effective concentrations (EC) of catechin, epicatechin, PCB1 and PCB2 were first determined in H_2_O_2_-treated HKC-8 cells, and the EC_50_ of these flavan-3-ols ranged from approximately 1 µM to 90 µM in the prevention of H_2_O_2_-induced cell death (measured by MTT assay) (Appendix A). Therefore, the effects of the catechins on hypoxic HKC-8 cells were investigated at the concentration of 100 µM. As shown in Figure 1, there was a minimal effect of the flavan-3-ols on cell viability of HKC-8 cells under normoxia conditions, but exposure to hypoxia (1% O_2_) significantly induced cell death, as indicated by the total positivity of Annexin-V and/or 7-AAD staining from 5.2 ± 0.4% under normoxia to 28.4 ± 1.45% under hypoxia (*p* < 0.0001, two-tailed *t*-test, n = 4). Addition of the flavan-3-ols significantly reduced the hypoxia-induced cell death to 11.2 ± 1.34% by catechin (Catechin vs. Medium, *p* < 0.0001, two-tailed *t*-test, n = 4), 17.3 ± 1.30% by epicatechin (Epicatechin vs. Medium, *p* < 0.0001, two-tailed *t*-test, n = 4), 19.1 ± 1.45% by PCB1 (PCB1 vs. Medium, *p* < 0.0001, two-tailed *t*-test, n = 4), and 20.0 ± 1.53% by PCB2 (PCB2 vs. Medium, *p* = 0.0009, two-tailed *t*-test, n = 4) (Figure 1). The cytoprotective activity of these flavan-3-ols ranged from high to low as follows: catechin > epicatechin (Catechin vs. Epicatechin, *p* = 0.0006) = PCB1 (Epicatechin vs. PCB1, *p* = 0.117) > PCB2 (PCB1 vs. PCB2, *p* = 0.0321) (Figure 1).

The interplay between hypoxia, mitochondrial dysfunction or diminished MMP, ROS generation and cell death has been well documented [20,21,22]. To further confirm the cytoprotection of flavan-3-ols, the effect of catechin on MMP and intracellular ROS in hypoxic HKC-8 cells was examined. As shown in Figure 2, the MMP was significantly decreased following exposure to hypoxia, as indicated by increased JC-1 red intensity from 4.80 ± 0.50% in normoxia to 40.80 ± 2.90% in hypoxia (*p* < 0.0001, two-tailed *t*-test, n = 3), and addition of catechin significantly prevented hypoxia-induced mitochondrial dysfunction, which was indicated by lowering the JC-1 red intensity in hypoxia to 17.90 ± 1.80% (Catechin vs. Medium, *p* = 0.0003, two-tailed *t*-test, n = 3). Similarly, Figure 3 shows an increase in intracellular ROS levels in hypoxic HKC-8 cells, indicated by an increase in DCF intensity in hypoxia (Hypoxia/Medium vs. Normoxia, *p* < 0.0001, two-tailed *t*-test, n = 3), which was inhibited by catechin (Catechin vs. Medium, *p* < 0.0001).

### 3.2. Up-Regulated TXNRD 1, a Main Node of the Catechin-Activated Network in Hypoxic HKC-8 Cells

Hypoxia activates oxidative stress that damages tissues including blood vessels and impairs tissue repair, exacerbating the hypoxia (chronic hypoxia) in the development of kidney diseases [3]. Further, HIF-1α is a key transcription factor that functions as a master regulator of cellular response to hypoxia in the kidney tubular epithelial cells [23,24]. To understand the molecular actions by which the catechin prevented hypoxia-induced mitochondrial dysfunction (loss of MMP), ROS generation and cell death in HKC-8 cells (Figure 1, Figure 2 and Figure 3), the expression of both HIF-1α and an oxidative stress-related gene expression profile (Appendix A) were examined in hypoxic HKC-8 cells in the absence or absence of the catechin. As expected, hypoxia prevented cellular HIF-1α degradation, as indicated by the increased levels of HIF-1α protein in HKC-8 cells in hypoxia/medium compared with those in normoxia, and incubation with catechin lowered the HIF-1α protein levels in hypoxia (Figure 4). The PCR array analysis showed that both up-regulation (≥2-fold) of 15 genes (*ALOX12*, *BNIP3*, *CYBB*, *CYGB*, *DUOX2, DUSP1*, *GPX2*, *MBL2*, *NCF2*, *NOX4*, *PTGS1*, *PTGS2*, *SEPP1*, *SPINK1*, and *TTN*) and down-regulation (≥2-fold) of 14 genes (*DHCR22*, *GCLC*, *HPRT1*, *HSPA1A*, *NOX5*, *NUDT1*, *PRNP*, *SRXN1*, *BAG2*, *GCLM*, *GLA*, *HSP90AA1*, *SLC7A11*, and *TXNRD1*) were seen in hypoxic HKC-8 cells compared with those in normoxia condition (Figure 5A). Further, treatment of hypoxic HKC-8 cells with catechin significantly upregulated (≥2-fold) 9 genes (*CCL5*, *GCLC*, *HSPA1A*, *NCF1*, *NOX5*, *PRDX3*, *PRDX4*, *PTGS2*, and *TXNRD1*) and downregulated (≥2-fold) 5 genes (*CYBB*, *DUSP1*, *NCF2*, *SOD3*, and *SRXN1*) compared with those untreated hypoxic cells (Figure 5B). Interestingly, the upregulated *CYBB* (NADPH oxidase 2, *NOX2*), *DUSP1* (Dual specificity phosphatase 1) and *NCF2* (NADPH oxidase activator 2) by hypoxia in HKC-8 cells in Figure 5A were down-regulated by the catechin treatment (Figure 5B), and at the same time, the down-regulated *GCLC* (Glutamate-cysteine ligase catalytic subunit for glutathione synthesis), *HSPA1A* (Heat shock 70 kDa protein 1A), SRXN1 (Sulfiredoxin 1) and *TXNRD1* (Thioredoxin reductase 1) by the hypoxia (Figure 5A) were up-regulated by the catechin treatment (Figure 5B).

The protein–protein interaction analysis of the affected genes in Figure 5 using STRING databases showed that the upregulated *CYBB*, *PTGS2*, *NOX4*, *DUOX2* and *GPX2* as well as downregulated *HSP90AA1* and *TXNRD1* were the central nodes that mostly linked to other nodes in the hypoxia-activated network in HKC-8 cells (Figure 6A), whereas the upregulated *TXNRD1* was a main node in the catechin-activated network (Figure 6B). These observations were further confirmed by the molecular function analysis, showing that both *DUOX2* and *NOX4* were associated with six (e.g., superoxide-generating NAD(P)H oxidase activity, NAD(P)H oxidase H_2_O_2_-forming activity, oxidoreductase activity, acting on NAD(P)H, and oxidoreductase activity) out of ten cellular functions, followed by *TXNRD1* in five functions (e.g., antioxidant activity, flavin adenine dinucleotide binding, and electron transfer activity) (Table 1). In catechin-treated hypoxic HKC-8 cells, up-regulated *TXNRD1* played a role in seven (e.g., antioxidant activity, oxidoreductase activity, flavin adenine dinucleotide binding, and electron transfer activity) of eleven total affected functions (Table 2).

To verify the regulation of TXNRD1 expression, a key molecule, in HKC-8 cells in response to hypoxia and catechin treatment, the TXNRD1 protein expression was examined in hypoxic HKC-8 cells in the absence or presence of the flavan-3-ols (catechin, epicatechin, PCB1, and PCB2). As shown in Figure 7, the level of TXNRD1 protein was significantly decreased in the hypoxic cells compared with that in normoxia, but it was significantly up-regulated by the treatment with the flavan-3-ols to the level close to that in normoxia condition.

### 3.3. Up-Regulated TXNRD1, a Cytoprotective Protein Mediating Cytoprotection of Catechin against Cell Death in Hypoxic HKC-8 Cells

To confirm the role of up-regulated TXNRD1 in the cytoprotection of the catechin in hypoxic HKC-8 cells, overexpression of this protein in HKC-8 cells (“TXNRD1-overexpressed phenotype”) was generated by stable expression of pHEX6300^TXNRD1^ plasmid (Appendix A). Like non-transfected HKC-8 cells, TXNRD1 protein levels were decreased by hypoxia in HKC-8^Control^ cells, which was also prevented by catechin treatment (Figure 8). In contrast, in HKC-8^TXNRD1^ cells, the expression of TXNRD1 protein was significantly downregulated by hypoxia as compared with that in the normoxia condition, but its levels were not significantly affected by the catechin treatment as they remained at high levels (Figure 8). This indicates that the “TXNRD1 overexpressed phenotype” of HKC-8^TXNRD1^ cells was not significantly changed in the hypoxia condition. Cell death analysis showed that compared with the control phenotype (HKC-8^Control^ cells), hypoxia-induced cell death was largely inhibited with TXNRD1 overexpression, which was evidenced by the fact that the cell death rate of 7.2 ± 1.0% in hypoxic HKC-8^TXNRD1^ cells was significantly lower than the rate of 36.2 ± 5.1% in hypoxic HKC-8^Control^ cells (*p* < 0.0001, two-tailed *t*-test, n = 4) (Figure 9). Catechin treatment also resulted in a significant decrease in cell death in both hypoxic HKC-8^Control^ (Catechin vs. Medium, *p* = 0.0001, two-tailed *t*-test, n = 4) and hypoxic HKC-8^TXNRD1^ cells (Catechin vs. Medium, *p* = 0.0003, two-tailed *t*-test, n = 4) (Figure 9). Further, the cell death in catechin-treated HKC-8^TXNRD1^ cells (2.8 ± 0.6%) was significantly lower than that (12.1 ± 2.0%) in catechin-treated HKC-8^Control^ cells (*p* = 0.0001, two-tailed *t*-test, n = 4). Statistical analysis with two-way ANOVA showed that the number of dead cells in the HKC-8^TXNRD1^ cells was significantly lower than in the HKC-8^Control^ cells in all of these conditions (*p* < 0.0001). These data imply that not only the overexpression of TXNRD1 in HKC-8^TXNRD1^ cells significantly protected HKC-8 cells from hypoxia-induced cell death, but also that the cytoprotective activity of flavan-3-ols (as shown in Figure 1) was mainly TXNRD1-dependent.

TRi-1 is a specific inhibitor of TXNRD1 [25,26]. To further confirm the role of TXNRD1 in the cytoprotection of the catechin in hypoxic HKC-8 cells, the effect of TRi-1 on the cell death in the hypoxic HKC-8 cells treated with TRi-1 was investigated. As expected, a prolonged hypoxia induced cell death from 6.4 ± 1.1% in normoxia to 35.3 ± 4.2% in hypoxia (*p* = 0.0003, two-tailed *t*-test, n = 3) that was reduced to 14.6 ± 2.3% with catechin (*p* = 0.0017, two-tailed *t*-test, n = 3) (Figure 10). The beneficial effect of the catechin on cell survival was diminished in the presence of TRi-1, as indicated by cell death of 17.5 ± 3.2% in cultures with 1 µM Tri-1, 36.3 ± 4.5% with 5 µM Tri-1, and 50.8 ± 5.3% with 10 µM Tri-1 (*p* < 0.0001, one-way ANOVA, n = 3) (Figure 10). These data suggest that specific inhibition of TXNRD1 using TRi-1 nullified the cytoprotection of the catechin against hypoxia in HKC-8 cells.

## 4. Discussion

The pro-oxidant ROS can be classified into two groups: one includes free radical ROS such as superoxide O_2_^•−^ and hydroxyl OH^•^, and the other includes non-radical molecules such as H_2_O_2_ that can be converted to radical ROS [27]. Within cells, ROS can be produced by enzymatic reactions such as NADPH oxidase in the endoplasmic reticulum and metabolic enzymes such as lipoxygenase, and xanthine oxidase in the cytosol or by nonenzymatic reactions such as from the mitochondrial respiratory chain [28]. Among all these different sources, the main ROS producers are mitochondria, where electron leakage from the site of the respiratory chain can react with O_2_, producing superoxide and be subsequently converted to other ROS molecules [28,29,30]. In normoxia conditions, these ROS molecules, the by-products from normal cellular metabolism, are mainly controlled or balanced by several antioxidant defense mechanisms including the activity of antioxidant enzymes (e.g., SOD, CAT and GPXs), redox systems such as the thioredoxin system (Trx–TrxR–NADPH), the glutaredoxin system (GLRX–GR–GSH] and peroxiredoxins; and non-enzyme antioxidant molecules such as reduced GSH and vitamins [31]. However, whether the antioxidant defense mechanisms are activated differently in different tissues or in response to different stimuli is not fully understood.

Oxidative stress is defined as the situation of excessive ROS accumulation due to an imbalance between ROS production and antioxidant defense activities as mentioned above, and it consequently results in cell and tissue damage [32,33]. Hypoxia primarily causes mitochondrial dysfunction that produces a ROS burst due to the modified electron transport chain [34,35]. Indeed, hypoxia induces MMP diminishment and ROS generation in hypoxic HKC-8 cells compared with controls in normoxia (Figure 2 and Figure 3). At the same time, it changes the expression of a panel of genes by activation of HIF-1 [36,37], for example, 630 genes in neuron cells [38] and 280 genes in endothelial cells after 16 h of hypoxia [39], to adjust the cellular metabolism to a low O_2_ environment. In most cells under normoxia, HIF-1α expresses at fundamental a low level, whereas under hypoxia, HIF-1α transcription is significantly upregulated [40], which is also seen in hypoxic HKC-8 cells (Figure 4). Following a prolonged period (72 h) of hypoxia exposure, the expression of 29 oxidative stress-related genes (15 up-regulated, 14 down-regulated) was affected compared with the normoxia condition in HKC-8 cells (Figure 5A), and a high number of HIF-1α (Q16665) binding sites were found in the DNA sequence of most of these genes (gene transcription regulation database: gtrd.biouml.org, accessed on 23 March 2022) [41] (Appendix A). Based on the number of HIF-1 binding sites, the up-regulation of *ALOX12*, *BNIP3*, *CYGB*, *DUSP1*, *MBL2*, *PTGS1*, and *TTN* and the down-regulation of *GCLC*, *HSPA1A*, *NOX5*, *PRNP*, *BAG2*, *HSP90AA1*, *SLC7A11*, and *TXNRD1* may be the main transcriptional targets of HIF-1. Interestingly, catechin suppresses hypoxia-stimulated HIF-1α expression (Figure 4), but its effect on these HIF-1α-regulated genes is only found in the expression of *HSPA1A*, *NOX5*, *TXNRD1*, *CYBB*, *DUSP1*, and *NCF2* (Figure 5B). Further, the catechin action is not associated with the number of HIF-1 binding sites of these genes (Appendix A). These data suggest that the molecular action of catechin in hypoxic HKC-8 cells may be mediated by both HIF-1α-dependent and -independent mechanisms. The network analysis shows that some key nodes were up-regulated including DUOX2 (ROS production), CYBB (NOX2, ROS production), PTGS2 (COX2, prostaglandin biosynthesis/inflammation), NOX4 (ROS production), and GPX2 (antioxidant), whereas others including HSP90AA1 (chaperone, prosurvival autophagy) and TXNRD1 (antioxidant, thioredoxin system) were down-regulated. These changes were correlated with the loss of MMP (Figure 2) and ROS production (Figure 3), resulting in the significant cell death (Figure 1). These data suggest that a prolonged exposure to hypoxia predominantly induces oxidative stress by both stimulation of ROS production, mainly from DUOX2, CYBB, PTGS2 and NOX2, and repression of the antioxidant thioredoxin system by down-regulation of TXNRD1 expression.

Both grape skin and seed extracts are rich in common flavan-3-ols-catechin monomers ((+)-catechin and (−)-epicatechin) and procyanidin dimers (PCB1 and PCB2) [9], and the catechin is the most abundant compared with others in most studies [9], suggesting that the catechin is a major flavan-3-ol in grape skin and seed extracts. Considering the chemistry, the hydroxyl groups on the B-ring of the catechins can interact with ROS or metal ions (e.g., Fe^2+^), resulting in direct antioxidant effects [42]. Based on the measurement of inhibition of ONOO^−^ mediated tyrosine nitration and free radical (DPPH, ROO) scavenging, the antioxidant activity of the catechin is the highest among these flavan-3-ol compounds tested [43,44], which are provided in Appendix A. Consistent with this evidence, the cytoprotection of the catechin by the reduction of H_2_O_2_− and hypoxia-induced cell death of cultured HKC-8 cells was also the highest in the present study (Figure 1 and Appendix A).

In addition to the direct mechanisms of scavenging ROS and chelating metal ions, catechins or flavan-3-ols have been found to regulate the expression and activity of antioxidant/prooxidant enzymes. This includes upregulation of antioxidant enzymes (e.g., SOD, GPX/GSH and CAT) and inhibition of prooxidant enzyme activity (e.g., NADPH oxidase, xanthine oxidase) as well as stress-related signaling pathways (e.g., TNF-α, and activator protein 1) in different experimental systems [42]. Whether the flavan-3-ols renders the same effects in hypoxic HKC-8 cells or in the kidney with chronic hypoxia requires further investigation. The PCR array showed that the catechin upregulated *CCL5*, *GCLC*, *HSPA1A*, *NCF1*, *NOX5*, *PRDX3*, *PRDX4*, *PTGS2* and *TXNRD1*, whereas it downregulated *CYBB*, *DUSP1*, *NCF2*, *SOD3*, and *SRXN1* in hypoxic HKC-8 cells (Figure 5B). A network analysis indicated that the upregulation of TXNRD1, an enzyme of the antioxidative thioredoxin system, played a key role in catechin action against the oxidative stress in hypoxic HKC-8 cells (Figure 6). Further studies of overexpressed TXNRD1 or its inhibition using TRi-1, a specific TXNRD1 inhibitor [25,26], might confirm that TXNRD1 at least in part mediated the cytoprotection of the catechin against hypoxia-induced cell death in human proximal tubular HKC-8 cells (Figure 9 and Figure 10).

## 5. Conclusions

Evidence in the literature suggests that chronic hypoxia that induces oxidative stress via mitochondrial dysfunction might be a key pathogenic factor for the development of CKD. Using grape-based supplements as dietary antioxidants, mainly found as flavan-3-ols (catechin, epicatechin, PCB1 and PCB2), to reduce the oxidative stress in diseased kidneys has received much attention recently. To our knowledge, our results for the first time demonstrate that flavan-3-ols, especially catechin, prevent hypoxia-induced cell death in human proximal tubular epithelial cells, mainly by prevention of hypoxia-induced down-regulation of endogenous antioxidant TXNRD1 expression. Although the molecular actions of the catechin in the regulation of the TXNRD1 expression in hypoxic HKC-8 cells require further investigation, catechin could be widely applied as an active compound or a promising candidate pro-drug for the design and development of novel pharmaceutical products against chronic hypoxia-related cell death or disturbance of kidney structure and function.

## Figures and Tables

**Figure 1 antioxidants-11-01399-f001:**
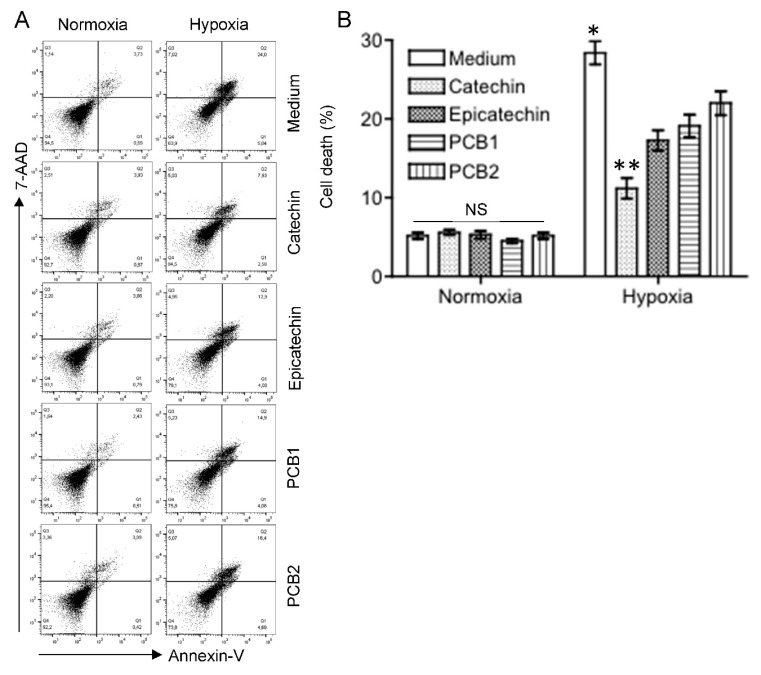
Different cytoprotective activities of flavan-3-ols in the reduction of hypoxia-induced cell death in HKC-8 cells. HKC-8 cells in 24-well plates were incubated in an atmosphere containing 1% O_2_ (hypoxia) or 20% O_2_ (normoxia) in the absence or presence of flavan-3-ols at 100 µM for 72 h. Cell death was measured using FACS analysis. (**A**) Typical FACS graphs showing the positive staining populations (Annexin-V: left lower quadrant—Q3 and left upper quadrant—Q2; 7-AAD: left upper quadrant—Q2 and upper right quadrant—Q1), (**B**) Data are presented as mean ± standard deviation (SD) of four determinants of total cell death (Q1–3) and are a representative of three separate experiments. NS: no significant difference between medium and any of flavan-3-ols, * *p* < 0.01 (Medium vs. any of flavan-3-ols, one-way ANOVA), ** *p* < 0.001 (Catechin vs. any of epicatechin, PCB1 and PCB2, one-way ANOVA).

**Figure 2 antioxidants-11-01399-f002:**
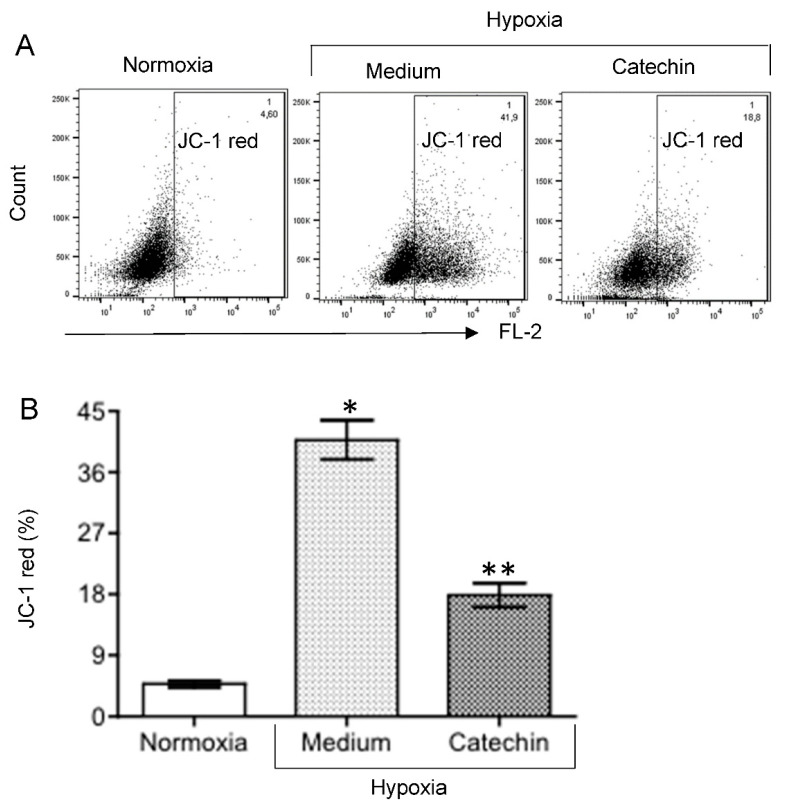
Effect of catechin on MMP in hypoxic HKC-8 cells. HKC-8 cells in 24-well plates were incubated in the atmosphere containing 1% O_2_ (hypoxia) or 20% O_2_ (normoxia) in the absence or presence of 100 µM catechin for 72 h. MMP was measured using FACS analysis with JC-1 staining. (**A**) Typical FACS graphs showing the degree of JC-1 aggregation as indicated by the intensity (%) of red fluorescence in the gated population. (**B**) Data are presented as mean ± SD of three measurements. FL-2: * *p* < 0.0001 (Medium vs. Normoxia, two-tailed *t*-test), ** *p* = 0.0003 (Catechin vs. Medium, two-tailed *t*-test).

**Figure 3 antioxidants-11-01399-f003:**
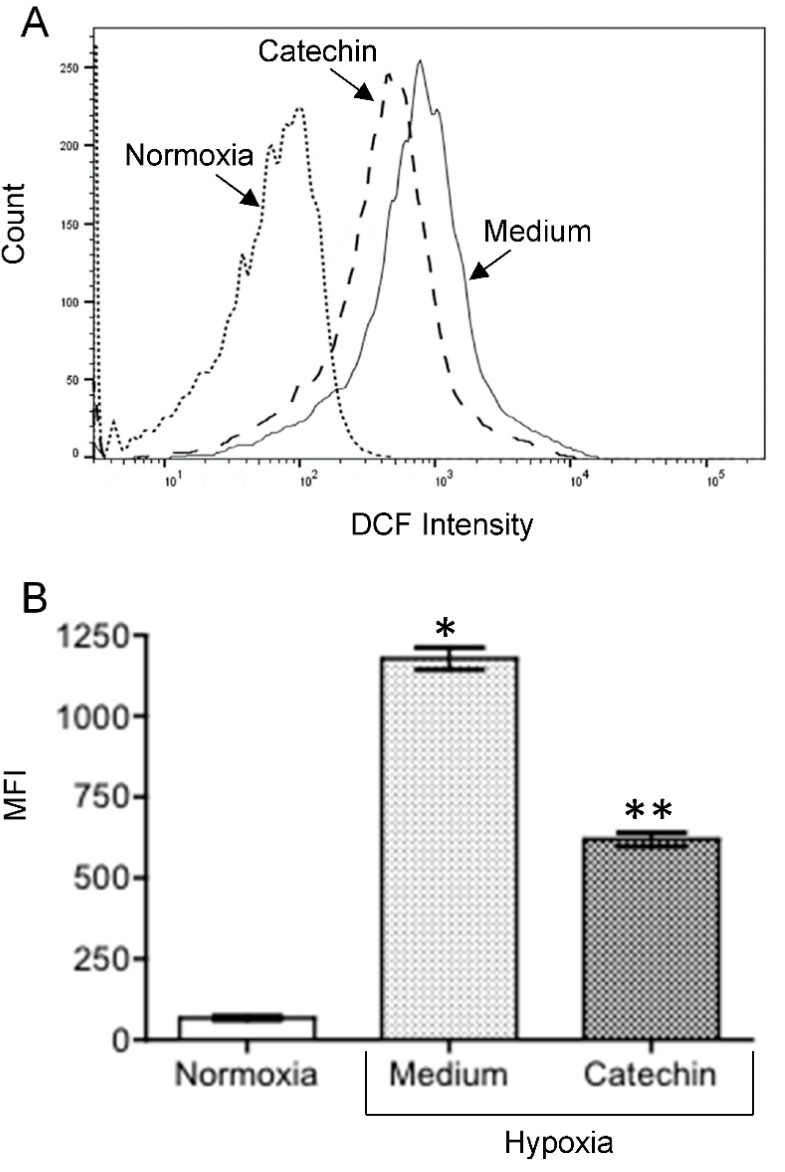
Effect of catechin on ROS levels in hypoxic HKC-8 cells. HKC-8 cells in 24-well plates were incubated in an atmosphere containing 1% O_2_ (hypoxia) or 20% O_2_ (normoxia) in the absence or presence of 100 µM catechin for 72 h. ROS was measured using FACS analysis with DCF-DA staining. (**A**) Typical FACS graphs showing the intensity of oxidized DCF fluorescence: dotted line—Normoxia; solid line;—hypoxia with medium; broken line—hypoxia with catechin. (**B**) Data of MFI (mean fluorescent intensity) are presented as mean ± SD of three determinants. * *p* < 0.0001 (Medium vs. Normoxia, two-tailed *t*-test), ** *p* < 0.0001 (Catechin vs. Medium, two-tailed *t*-test).

**Figure 4 antioxidants-11-01399-f004:**
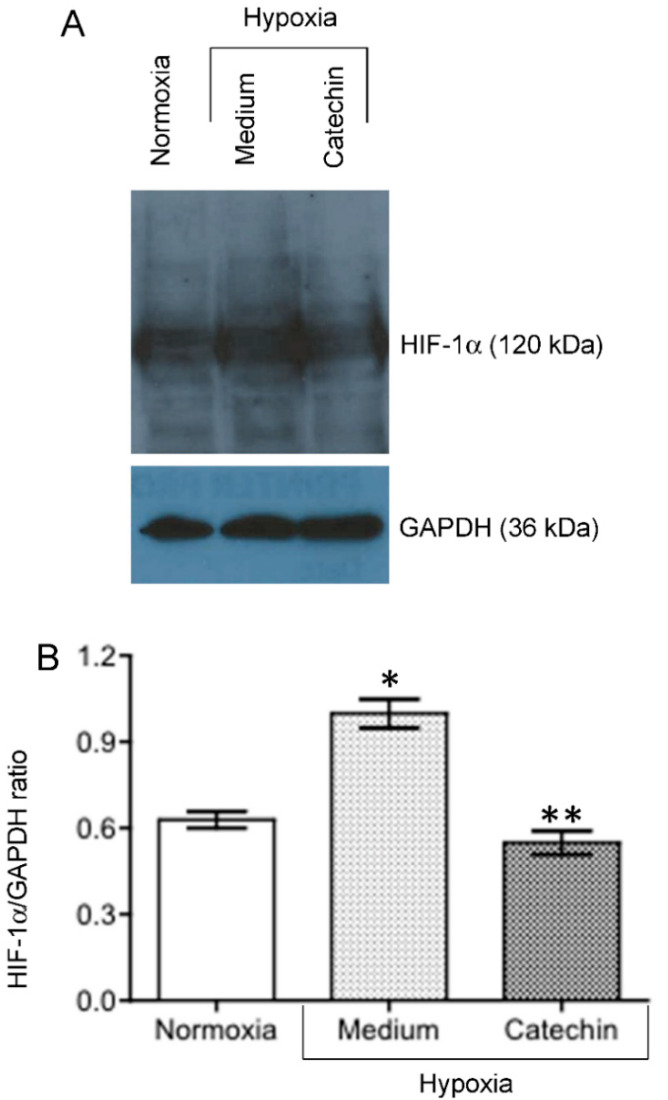
Effect of catechin on expression of HIF-1α protein in hypoxic HKC-8 cells. HKC-8 cells in plastic petri dishes were incubated in an atmosphere containing 1% O_2_ (hypoxia) or 20% O_2_ (normoxia) in the absence or presence of 100 µM catechin for 24 h. HIF-1α protein levels in cellular protein extracts were semi-quantitatively determined using Western blot analysis. (**A**) A typical blot of both HIF-1α and GAPDH protein of two separate experiments. (**B**) HIF-1α/GAPDH ratios in each group. * *p* = 0.0122 (Medium vs. Normoxia, two-tailed *t*-test), ** *p* = 0.01 (Medium vs. Catechin, two-tailed *t*-test).

**Figure 5 antioxidants-11-01399-f005:**
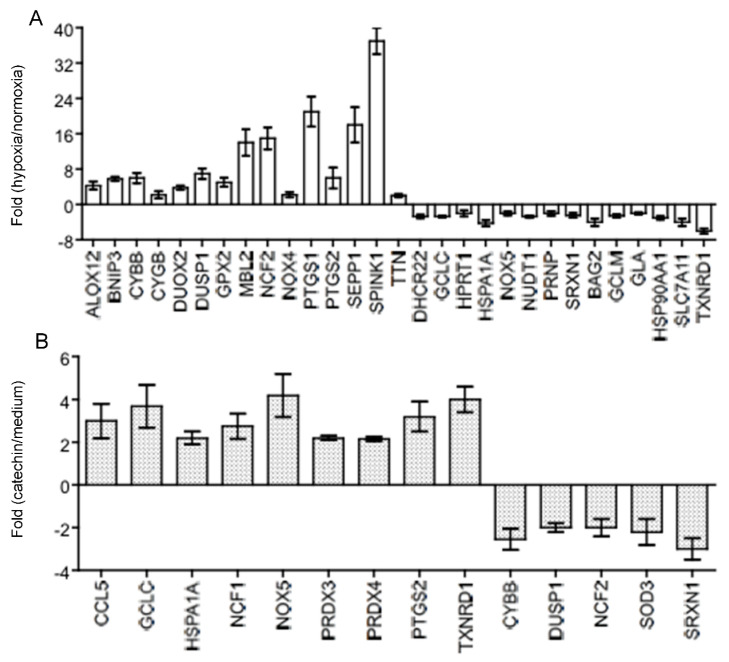
The expression of oxidative stress-related genes in hypoxic HKC-8 cells in the absence or presence of 100 µM of catechin. After 72 h of incubation, the gene expression was determined using PCR array analysis compared with the expression of housekeeping genes (Appendix A). (**A**) Fold changes of gene expression in hypoxia compared with in normoxia, (**B**) Fold changes of gene expression in catechin-treated hypoxic cells compared with in untreated (medium) hypoxic cells. These genes were chosen as their differences between hypoxia and normoxia or catechin-treated and medium were ≥2-fold and statistically significant (*p* < 0.05, two-tailed *t*-test, n = 3). This experiment was repeated three times in triplicate using independently prepared cDNAs, and the results were exhibited in almost identical patterns.

**Figure 6 antioxidants-11-01399-f006:**
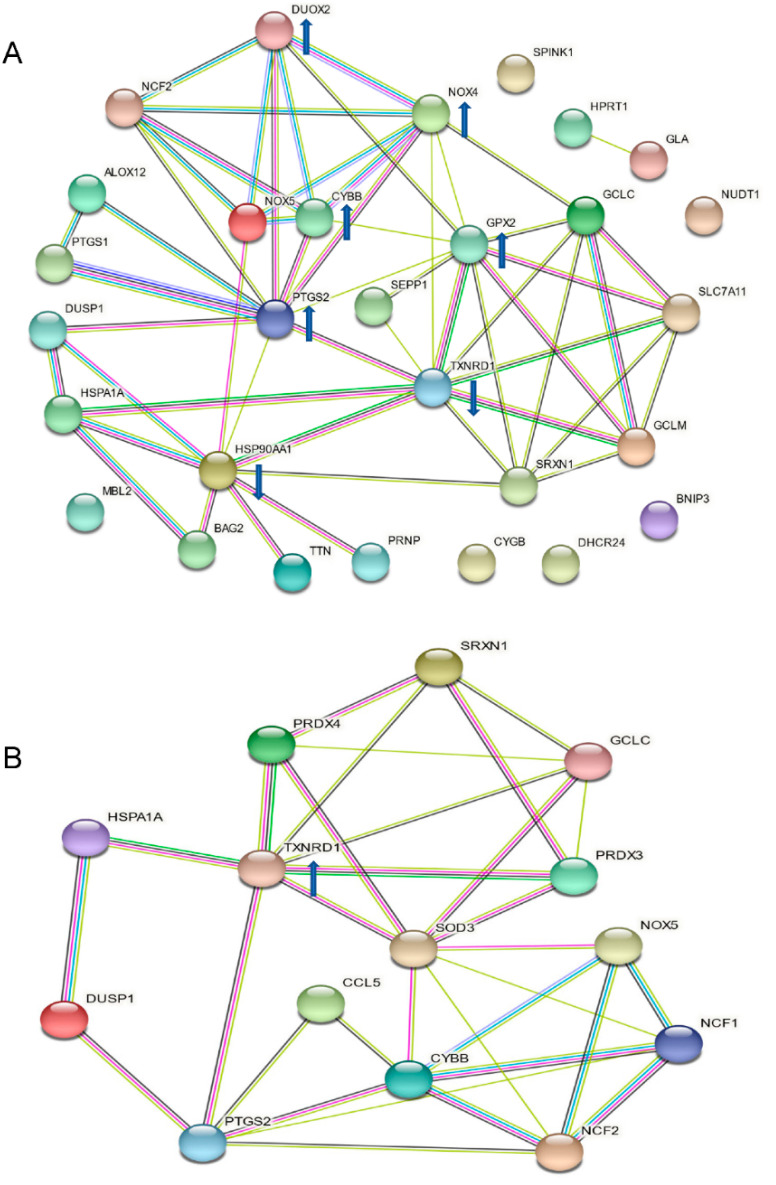
Interaction networks of affected oxidative stress-related molecules in HKC-8 cells in hypoxia or treated with catechin. The interaction networks of genes in Figure 2 were analyzed using STRING databases: (**A**) the interaction network of oxidative stress-related molecules in hypoxic HKC-8 cells; (**B**) the interaction network of oxidative stress molecules affected by catechin treatment in hypoxic HKC-8 cells. Known interactions: green line (from curated databases) and purple line (experimentally determined), predicted interactions: lime line (gene neighborhood), red line (gene fusion) and blue line (gene co-occurrence), and others: yellow line (textmining), indigo line (co-expression) and grey line (protein homology). Arrow pointing up: upregulated, arrow pointing down: downregulated.

**Figure 7 antioxidants-11-01399-f007:**
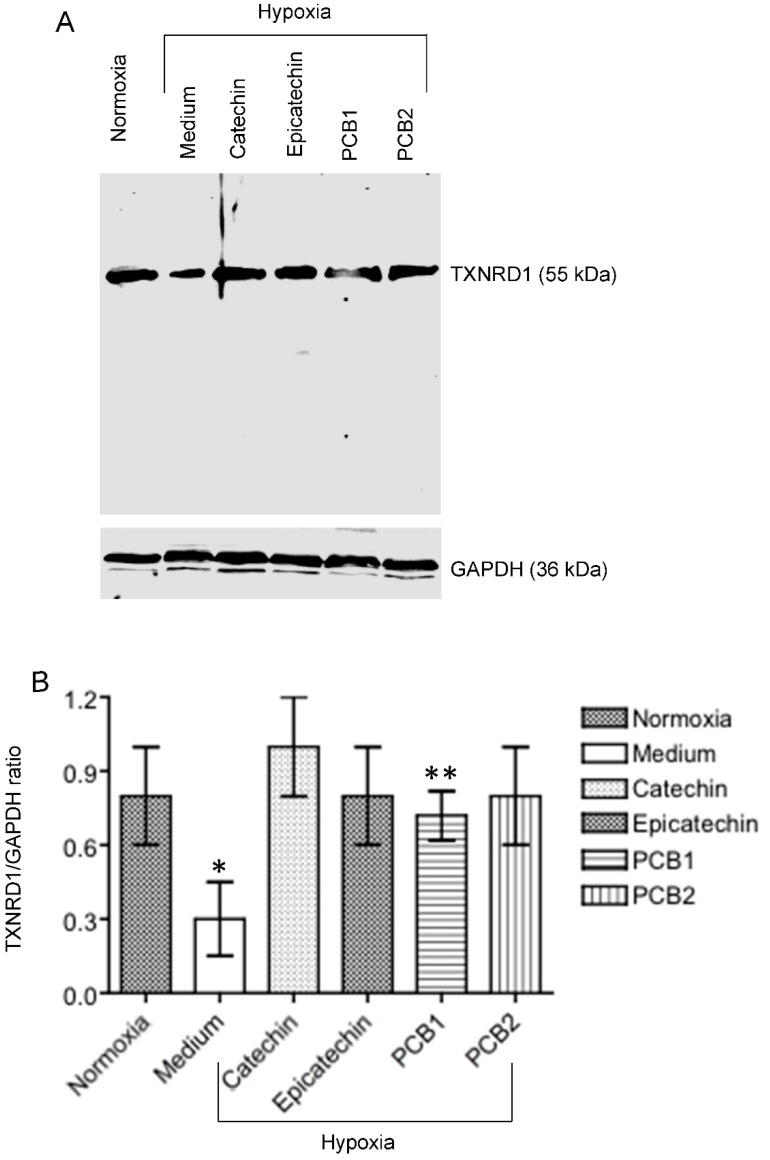
Regulation of TXNRD1 protein expression by hypoxia and flavan-3-ol treatment in HKC-8 cells. Protein samples were prepared from cells that were grown in normoxia or hypoxia in the absence (medium) or presence of 100 µM of catechin, epicatechin, PCB1 or PCB2 for 72 h. The levels of TXNRD1 protein and GAPDH were semi-quantitatively determined using Western blot analysis. (**A**) A typical blot of both TXNRDR1 and GAPDH protein of three separate experiments, (**B**) TXNRD1/GAPDH ratios in each group. * *p* < 0.05 (Medium vs. any of Normoxia, Catechin, Epicatechin and PCB2), ** *p* > 0.05 (Medium vs. PCB1) (One-way ANOVA, n = 3).

**Figure 8 antioxidants-11-01399-f008:**
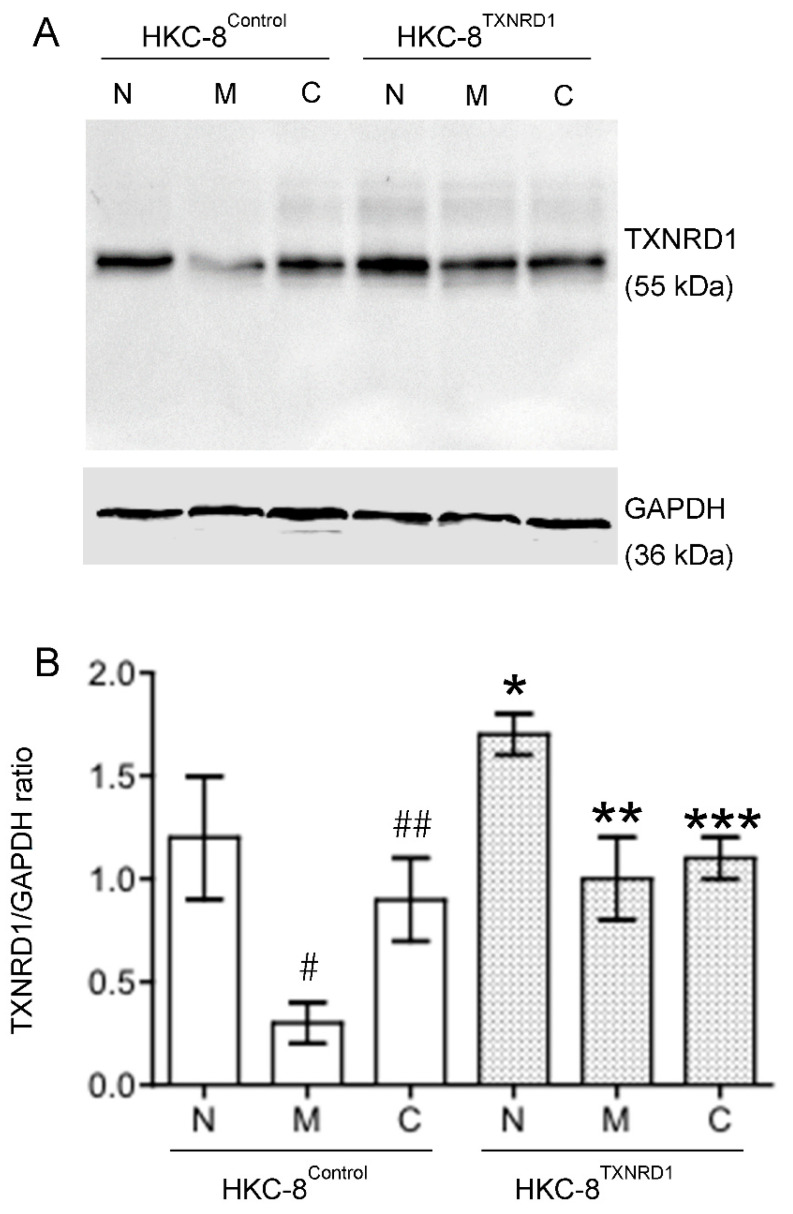
Regulation of TXNRD1 protein expression by hypoxia and flavan-3-ol treatment in in both HKC-8^TXNRD1^ and HKC-8^Control^ cells. Protein samples were prepared from both types of cells that were grown in normoxia or hypoxia in the absence (medium) or presence of 100 µM of catechin for 72 h. The levels of TXNRD1 protein and GAPDH were semi-quantitatively determined using Western blot analysis. (**A**) A typical blot of both TXNRDR1 and GAPDH protein of three separate experiments, (**B**) TXNRD1/GAPDH ratios in each group. N: Normoxia, M: Hypoxia with Medium, C: Hypoxia with Catechin. * *p* < 0.05 (HKC-8^TXNRD1^ vs. HKC-8^Control^ cells in normoxia), ** *p* > 0.05 (HKC-8^TXNRD1^ in hypoxia vs. HKC-8^Control^ cells in normoxia), *** *p* > 0.05 (Hypoxic HKC-8^TXNRD1^ with Catechin vs. Hypoxic HKC-8^TXNRD1^ with Medium) (One-way ANOVA, n = 3). # *p* = 0.0079 (HKC-8^Control^ cells: Normoxia vs. Hypoxia/Medium), ## *p* = 0.0097 (HKC-8^Control^ cells: Medium vs. Catechin) (two-tailed *t*-test).

**Figure 9 antioxidants-11-01399-f009:**
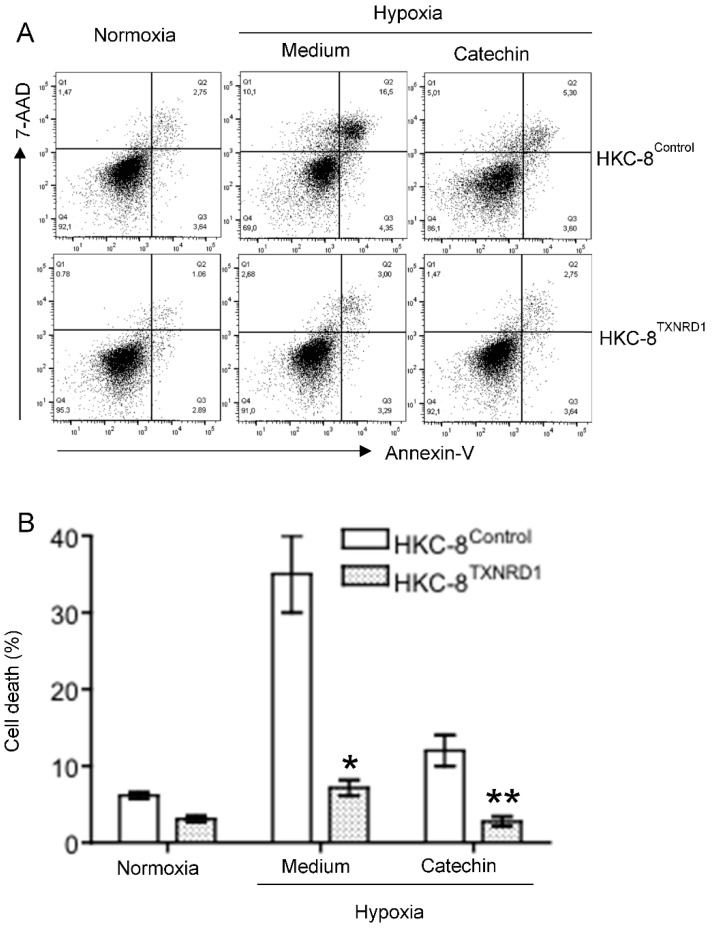
Cytoprotection of TXNRD1 overexpression against hypoxia-induced cell death in HKC-8 cells. Both HKC-8^Control^ and HKC-8^TXNRD1^ cells in 24-well plates were incubated in an atmosphere containing 1% O_2_ (hypoxia) or 20% O_2_ (normoxia) in the absence (medium) or presence of 100 µM catechin for 72 h. Cell death was measured using FACS analysis. (**A**) Typical FACS graphs showing the dead cell populations (Q1–Q3), (**B**) Data are presented as mean ± SD of four measurements of total cell death and are representative of three separate experiments. * *p* < 0.0001 (hypoxic HKC-8^TXNRD1^ vs. hypoxic HKC-8^Control^ in Medium, two-tailed *t*-test, n = 4), ** *p* = 0.0001 (hypoxic HKC-8^TXNRD1^ vs. hypoxic HKC-8^Control^ in the presence of catechin, two-tailed *t*-test, n = 4).

**Figure 10 antioxidants-11-01399-f010:**
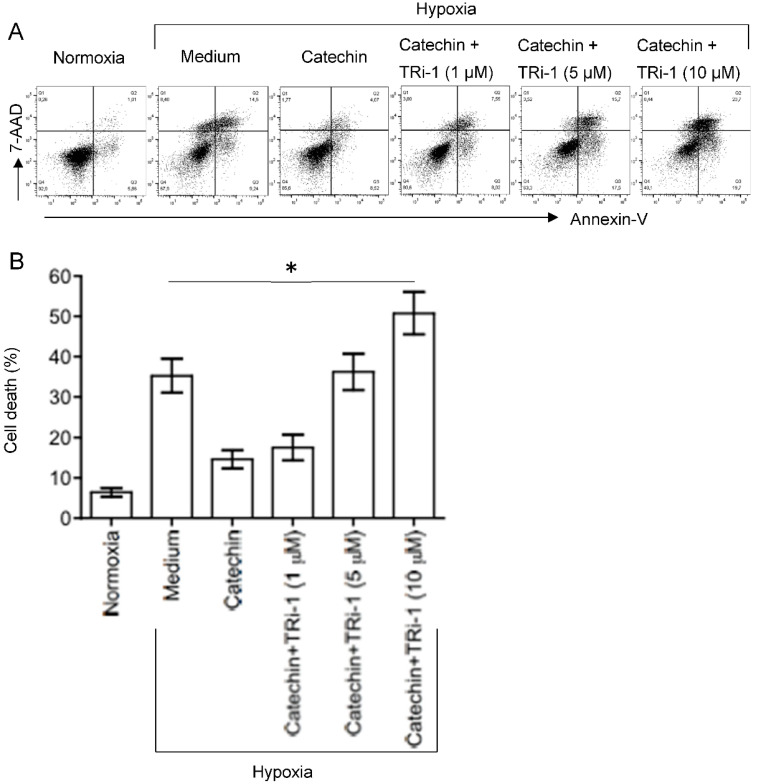
Inactivation of catechin cytoprotection against hypoxia-induced cell death in HKC-8 cells. HKC-8 cells in 24-well plates were incubated in an atmosphere containing 1% O_2_ (hypoxia) or 20% O_2_ (normoxia) in the absence (medium) or presence of 100 µM catechin with or without TRi-1 for 72 h. Cell death was measured using FACS analysis. (**A**) Typical FACS graphs showing the dead cell populations (Q1–Q3), (**B**) Data are presented as mean ± SD of four measurements of total cell death and are a representative of three separate experiments. * One-way ANOVA: *p* < 0.01 (Medium vs. Catechin, Medium vs. Catechin + 1 µM TRi-1), *p* > 0.05 (Medium vs. Catechin + 5 µM TRi-1), and *p* < 0.01 (Medium vs. Catechin + 10 µM TRi-1).

**Table 1 antioxidants-11-01399-t001:** Molecular functions activated with strength > 1 in hypoxic HKC-8 cells.

GO-Term	Description of Functions	Count in Network	Strength	False Discovery Rate
0004666	Prostaglandin-endoperoxide synthase activity	2 (*PTGS1*, *PTGS2*) of 2	2.83	0.0032
0004357	Glutamate-cysteine ligase activity	2 (*GCLC*, *GCLM*) of 2	2.83	0.0032
0016175	Superoxide-generating NAD(P)H oxidase activity	5 (*NCF2*, *DUOX2*, *NOX4*, *NOX5*, *CYBB*) of 11	2.49	3.54 × 10^−8^
0016174	NAD(P)H oxidase H_2_O_2_-forming activity	2 (*DUOX2*, *NOX4*) of 9	2.18	0.0197
0004601	Peroxidase activity	5 (*PTGS1*, *PTGS2*, *DUOX2*, *GPX2*, *CYGB*) of 41	1.92	3.00 × 10^−6^
0016209	Antioxidant activity	7 (*PTGS1*, *PTGS2*, *DUOX2*, *GPX2*, *TXNRD1*, *SRXN1*, *CYGB*) of 74	1.8	3.54 × 10^−8^
0050660	Flavin adenine dinucleotide binding	5 (*NOX4*, *NOX5*, *CYBB*, *TXNRD1*, *DHCR24*) of 82	1.61	4.94 × 10^−5^
0016651	Oxidoreductase activity, acting on NAD(P)H	6 (*NCF2*, *DUOX2*, *NOX4*, *NOX5*, *CYBB*, *TXNRD1*) of 107	1.58	4.93 × 10^−6^
0009055	Electron transfer activity	4 (*NCF2*, *NOX4*, *GPX2*, *TXNRD1*) of 103	1.42	0.0039
0016491	Oxidoreductase activity	13 (*PTGS1*, *PTGS2*, *ALOX12*, *NCF2*, *DUOX2*, *NOX4*, *NOX5*, *CYBB*, *GPX2*, *TXNRD1*, *SRXN1*, *CYGB*, *DHCR24*) of 726	1.08	3.54 × 10^−8^

**Table 2 antioxidants-11-01399-t002:** Molecular functions activated with strength > 1 by catechin in hypoxic HKC-8 cells.

GO-Term	Description of Functions	Count in Network	Strength	False Discovery Rate
0008379	Thioredoxin peroxidase activity	2 (*PRDX3*, *PRDX4*) of 5	2.75	0.0036
0016175	Superoxide-generating NAD(P)H oxidase activity	4 (*NCF1*, *NCF2*, *NOX5*, *CYBB*) of 11	2.71	1.80 × 10^−7^
0016176	Superoxide-generating NADPH oxidase activator activity	2 (*NCF1. NCF2*) of 9	2.49	0.0070
0016668	Oxidoreductase activity, acting on a sulfur group of donors	2 (*TXNRD1*, *PRDX3*) of 12	2.37	0.0099
0016209	Antioxidant activity	6 (*PRDX4*, *TXNRD1*, *SRXN1*, *PRDX3*, *SOD3*, *PTGS2*) of 74	2.05	1.82 × 10^−8^
0004601	Peroxidase activity	3 (*PRDX3*, *PRDX4*, *PTGS2*) of 41	2.01	0.0018
0016651	Oxidoreductase activity, acting on NAD(P)H	6 (*TXNRD1*, *PRDX3*, *NCF1*, *NCF2*, *NOX5*, *CYBB*) of 107	1.89	1.01 × 10^−7^
0016667	Oxidoreductase activity, acting on a sulfur group of donors	3 (*TXNRD1*, *PRDX3*, *SRXN1*) of 59	1.85	0.0036
0050660	Flavin adenine dinucleotide binding	3 (*TXNRD1*, *CYBB*, *NOX5*) of 82	1.71	0.0070
0009055	Electron transfer activity	3 (*TXNRD1*, *NCF1*, *NCF2*) of 103	1.61	0.0116
0016491	Oxidoreductase activity	10 (*TXNRD1*, *PRDX3*, *PRDX4*, *SRXN1*, *SOD3*, *NOX5*, *NCF1*, *NCF2*, *CYBB*, *PTGS2*) of 726	1.28	1.50 × 10^−8^

Data were generated using STRIP database software (version 11.5, string-db.org, accessed on 23 March 2022; ELIXIR, Hinxton, Cambridgeshire, UK).

## Data Availability

All data are included in the manuscript.

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
