# Peer review of "Upregulation of Thioredoxin Reductase 1 Expression by Flavan-3-Ols Protects Human Kidney Proximal Tubular Cells from Hypoxia-Induced Cell Death"

_antioxidants, 2022, doi:10.3390/antiox11071399_

Round 1

Reviewer 1 Report

The manuscript by Zhu et al., describes the protective role of catechin on hypoxia-induced death by upregulating the thioredoxin reductase 1. Protective effect of catechins is well known from several years in various cell lines/in vivomodels and also in ROS-induced cell death. This manuscript potentially deals with the upregulation of TXNRD1 by catechins in hypoxia-induced cell death in renal tubular epithelial cells by using HKC-8 cell line. The data presented appears to be thorough and the conclusions drawn from the manuscript are justified. I have few concerns that are summarized below before publication is possible.

  1. Did authors do any free radical scavenging activity assay for the flavonols used in this study by comparing it with standard such as quercetin?
  2. Did authors do any mitochondrial assay, since manuscript is dealing with hypoxia which directly is related to mitochondrial dysfunctions.
  3. Since authors have used FACS analysis I would suggest them to quantify the intracellular ROS using DCF.
  4. It would be nice to show the HIF-1/2 ⍺ expression either by Western or by qPCR assay to further strengthen the conclusion drawn from the present study.
  5. Provide a better Western in the figure 4A and mention hypoxia from 2nd to 6th
  6. In figure 5B statistical analysis is missing in HKC-8control, compare it with the first bar (normoxia).
  7. Line 35 expand O2 on its first appearance in the introduction.
  8. In line 58 authors should expand on the “some kidney functions”

Author Response

We thanks the constructive comments from Reviewer #1. Here are the point-to point responses from us:

The manuscript by Zhu et al., describes the protective role of catechin on hypoxia-induced death by upregulating the thioredoxin reductase 1. Protective effect of catechins is well known from several years in various cell lines/in vivomodels and also in ROS-induced cell death. This manuscript potentially deals with the upregulation of TXNRD1 by catechins in hypoxia-induced cell death in renal tubular epithelial cells by using HKC-8 cell line. The data presented appears to be thorough and the conclusions drawn from the manuscript are justified. I have few concerns that are summarized below before publication is possible.

Q: Did authors do any free radical scavenging activity assay for the flavonols used in this study by comparing it with standard such as quercetin

RESPONSE: Although these compounds are known as strong antioxidants, we measured the inhibition of DPPH reaction – free radical scavenging activity by the remaining catechin, epicatechin, PCB1 and PCB2 compounds in the lab. Data were included as Suppl material Figure S3

Q: Did authors do any mitochondrial assay, since manuscript is dealing with hypoxia which directly is related to mitochondrial dysfunctions.

RESPONSE: We added this experiment showing the effect of catechin on hypoxia-induced mitochondrial membrane potential using JC-1 staining. Data were included as new Figure 2.

Q: Since authors have used FACS analysis I would suggest them to quantify the intracellular ROS using DCF.

RESPONSE: We added this experiment showing the effect of catechin on hypoxia-stimulated ROS production using DCF-DA staining. Data were included as new Figure 3

Q: It would be nice to show the HIF-1/2 ⍺ expression either by Western or by qPCR assay to further strengthen the conclusion drawn from the present study.

RESPONSE: We added this experiment showing the effect of catechin on HIF-1alpha expression in HKC-8 cells in in response to hypoxia. Data were included as new Figure 4.

Q: Provide a better Western in the figure 4A and mention hypoxia from 2nd to 6th

RESPONSE: The experiment were repeated 3 times – the image we presented in the manuscript was a representative of the analyses (new Figure 7). Other data in Figure 8 (Western blot) and Figure 6B (mRNA) also supported this result – catechin up-regulated TXNRD1 expression in hypoxic HKC-8 cells.

Q: In figure 5B statistical analysis is missing in HKC-8control, compare it with the first bar (normoxia).

RESPONSE: We added the p values in the Figure 8.

Q: Line 35 expand O2 on its first appearance in the introduction.

RESPONSE: It was corrected (please see line 40).

Q: In line 58 authors should expand on the “some kidney functions”

RESPONSE: It was revised (please see line 63 – 64).

Reviewer 2 Report

In this article the authors explore effects of flavanols in prevention of oxidative stress caused by hypoxia in HKC-8 cells

The article is well written and understandable. Experiments described and performed give enough data for the given conclusions. The use of English language is adequate and conforms to scientific English language, although there are some minor misspellings.

I only have a few minor suggestions for the authors listed below.

Line 66 please explain in short what complete K1 +/+ medium is

Line 90 please note that explanations of quadrants are referred to image 1.

Image 1B and 4 please adjust asterisk location on the image

Please adjust image locations in the text for them to be right after they are mentioned in the results section, and not all at the end of results section.

Line 250 please remove ,.

Line 430 please reformat sentence, it is confusing

Please refrain from providing speculations about results (i.e., line 218, and 230) in results section, these sentences belong to discussion or conclusion sections.

Author Response

Thank you for the constructive comments from Reviewer #2. Our point-to-point responses are listed below:

Comments

In this article the authors explore effects of flavanols in prevention of oxidative stress caused by hypoxia in HKC-8 cells

The article is well written and understandable. Experiments described and performed give enough data for the given conclusions. The use of English language is adequate and conforms to scientific English language, although there are some minor misspellings.

I only have a few minor suggestions for the authors listed below.

Line 66 please explain in short what complete K1 +/+ medium is

RESPONSE: It was included (please see line 76 – 79).

Line 90 please note that explanations of quadrants are referred to image 1.

RESPONSE: It was added (please see line 103 – 107)

Image 1B and 4 please adjust asterisk location on the image

RESPONSE: It was corrected

Please adjust image locations in the text for them to be right after they are mentioned in the results section, and not all at the end of results section.

RESPONSE: It was corrected.

Line 250 please remove ,.

RESPONSE: It was corrected.

Line 430 please reformat sentence, it is confusing

RESPONSE: It was revised

Please refrain from providing speculations about results (i.e., line 218, and 230) in results section, these sentences belong to discussion or conclusion sections.

RESPONSE: It was revised.

Round 2

Reviewer 1 Report

The authors have adequately addressed all of my previous concerns to my satisfaction. The current version of the manuscript is significantly improved.

I would suggest authors to include FACS trace curve legend in the figure itself for Fig. 3A.

Minor typos are found which could be fixed during proof reading.

Author Response

The authors have adequately addressed all of my previous concerns to my satisfaction. The current version of the manuscript is significantly improved.

I would suggest authors to include FACS trace curve legend in the figure itself for Fig. 3A.

Minor typos are found which could be fixed during proof reading.

RESPONSE: The legend was added in Figure 3A. Thanks for pointing out the error. The figure is more accessible for readers